# Gaussian Gated Linear Networks

**David Budden**[*]  **Adam H. Marblestone**[*]  **Eren Sezener**[*]
**Tor Lattimore**  **Greg Wayne**[†]  **Joel Veness**[†]

DeepMind

`aixi@google.com`

## Abstract

We propose the Gaussian Gated Linear Network (G-GLN), an extension to the recently proposed GLN family of deep neural networks. Instead of using back-propagation to learn features, GLNs have a distributed and local credit assignment mechanism based on optimizing a convex objective. This gives rise to many desirable properties including universality, data-efficient online learning, trivial interpretability and robustness to catastrophic forgetting. We extend the GLN framework from classification to multiple regression and density modelling by generalizing geometric mixing to a product of Gaussian densities. The G-GLN achieves competitive or state-of-the-art performance on several univariate and multivariate regression benchmarks, and we demonstrate its applicability to practical tasks including online contextual bandits and density estimation via denoising.

## 1 Introduction

Recent studies have demonstrated that backpropagation-free deep learning, particularly the Gated Linear Network (GLN) family [1, 2, 3], can yield surprisingly powerful models for solving classification tasks. This is particularly true in the online regime where data efficiency is paramount. In this paper we extend GLNs to model real-valued and multi-dimensional data, and demonstrate that their theoretical and empirical advantages apply to far broader domains than previously anticipated.

The distinguishing feature of a GLN is distributed and local credit assignment. A GLN associates a separate convex loss to each neuron such that all neurons (1) predict the target distribution directly, and (2) are optimized locally using online gradient descent. A half-space "context function" is applied per neuron to select which weights to apply as a function of the input features, allowing the GLN to learn highly nonlinear functions. This architecture gives rise to many desirable properties previously shown in a classification setting: (1) trivial interpretability given its piecewise linear structure, (2) exceptional robustness to catastrophic forgetting, and (3) provably universal learning; a sufficiently large GLN can model any well-behaved, compactly supported density function to any accuracy, and any no-regret convex optimization method will converge to the correct solution given enough data.

**Related Work.** We extend the previous Bernoulli GLN (B-GLN) formulation to model multivariate, real-valued data by reformulating the GLN neuron as a gated product of Gaussians. This Gaussian Gated Linear Network (G-GLN) formulation exploits the fact that exponential family densities are closed under multiplication [4], a property that has seen much use in Gaussian Process and related literature [5, 6, 7, 8, 9]. Similar to the B-GLN, every neuron in our G-GLN directly predicts the target distribution. This idea is shared with work in supervised learning where targets are predicted from intermediate layers. The motivations for local, layer-specific training include improving gradient propagation and representation learning [10, 11, 12, 13], decoding for representation analysis [14] and making neural networks more biologically plausible [15, 16, 17] by avoiding backpropagation. The use of context-dependent weight selection (gating) in the GLN algorithm family resembles proposals to improve the continual and multi-task learning properties of deep networks [18, 19, 20, 21, 22] by using a conditioning network to gate a principal network solving the task.

---

[*][†] Equal contributions.

**Paper Outline.** We begin by reviewing some background on weighted products of Gaussian densities, and describe how the relevant weights can be adapted using well-known online convex programming techniques [23]. We next show how to augment this adaptive form with a gating mechanism, inspired by earlier work on classification with GLNs [2, 3], which gives rise to the notion of neuron in G-GLNs. We then introduce G-GLNs, feed-forward networks of locally trained neurons, each computing a weighted product of Gaussians with input-dependent, gated weights. We conclude by providing a comprehensive set of experimental results demonstrating the impressive performance of the G-GLN algorithm across a diverse set of regression benchmarks and practical applications including contextual bandits and image denoising.

## 2 Background

The Gaussian distribution has a number of well-known properties that make it well suited for machine learning applications. Here we briefly review two of these important properties: closure under multiplication and convexity with respect to its parameters under the logarithmic loss, which we will later exploit to define our notion of a G-GLN neuron.

### 2.1 Weighted Products of Gaussian Densities

A weighted product of Gaussians is closed in the sense that it yields another Gaussian. More formally, let $\mathbb{R}_+$ denote the set of non-negative real numbers. For notational simplicity, we first construct the univariate case. Let $\mathcal{N}(\mu, \sigma^2)$ denote the univariate Gaussian PDF with mean $\mu \in \mathbb{R}$ and standard deviation $\sigma \in \mathbb{R}_+$. Now, given $m$ univariate Gaussian experts of the form $\mathcal{N}(\mu_1, \sigma_1^2)$, $\ldots, \mathcal{N}(\mu_m, \sigma_m^2)$ with associated PDFs

$$f_i(y) = \frac{1}{\sigma_i \sqrt{2\pi}} \exp\left\{ -\frac{1}{2} \left( \frac{y - \mu_i}{\sigma_i} \right)^2 \right\}, \tag{1}$$

and an $m$-dimensional vector of weights $w = (w_1, w_2, \ldots, w_m) \subset \mathbb{R}_+^m$, we define a weighted Product of Gaussians (PoG) as

$$\mathrm{PoG}_w(y\,;\, f_1(\cdot), \ldots, f_m(\cdot)) := \frac{1}{Z} \prod_{i=1}^{m} [f_i(y)]^{w_i} \quad \text{with} \quad Z := \int \prod_{i=1}^{m} [f_i(y)]^{w_i}\, dy. \tag{2}$$

It is straightforward to show that this formulation gives rise to a Gaussian distribution whose mean and variance jointly depend on $w$; see Appendix A for a short derivation. In particular we can exactly interpret the weighted product of experts as another Gaussian expert $\mathcal{N}\left(\mu_{\mathrm{PoG}}(w), \sigma_{\mathrm{PoG}}^2(w)\right)$ where

$$\sigma_{\mathrm{PoG}}^2(w) := \left[ \sum_{i=1}^{m} \frac{w_i}{\sigma_i^2} \right]^{-1} \quad \text{and} \quad \mu_{\mathrm{PoG}}(w) := \sigma_{\mathrm{PoG}}^2(w) \left[ \sum_{i=1}^{m} \frac{w_i \mu_i}{\sigma_i^2} \right]. \tag{3}$$

The same closure property holds for the multivariate case (e.g. see [24]). Let $\mathcal{N}(\mu, \Sigma)$ denote the $d$-dimensional multivariate Gaussian PDF, with mean $\mu \in \mathbb{R}^d$ and covariance matrix $\Sigma \in \mathbb{R}^{d \times d}$, and let $\mathcal{I}_d$ denote the $d$-dimensional identity matrix. In the general case, given $m$ multivariate $d$-dimensional Gaussian experts, $\mathcal{N}(\mu_1, \Sigma_1), \ldots, \mathcal{N}(\mu_m, \Sigma_m)$, we have

$$\Sigma_{\mathrm{PoG}}^{-1}(w) = \sum_{i=1}^{m} w_i \Sigma_i^{-1} \quad \text{and} \quad \mu_{\mathrm{PoG}}(w) = \Sigma_{\mathrm{PoG}}(w) \sum_{i=1}^{m} w_i \Sigma_i^{-1} \mu_i. \tag{4}$$

Note that $\mu_{\mathrm{PoG}}(w)$ is a convex combination of the means $\mu_i$ of its inputs, which implies that $\mu_{\mathrm{PoG}}(w)$ must lie within the convex hull formed from all the $\mu_i$. In the isotropic case with $\Sigma_i^{-1} = \tau_i \mathcal{I}_d$ for precision $\tau_i > 0$, Equation 4 simplifies to

$$\Sigma_{\mathrm{PoG}}^{-1}(w) = \left( \sum_{i=1}^{m} w_i \tau_i \right) \mathcal{I}_d \quad \text{and} \quad \mu_{\mathrm{PoG}}(w) = \left( \sum_{i=1}^{m} w_i \tau_i \right)^{-1} \sum_{i=1}^{m} w_i \tau_i \mu_i. \tag{5}$$

Note that if all the initial experts are isotropic, the product of Gaussians must also be isotropic. Although less general, the isotropic form has considerable computational advantages for high-dimensional multivariate regression (since the inverses can be computed in $\mathcal{O}(d)$ time), and will be used in our larger scale multivariate regression experiments.

## 2.2 Online Convex Programming Formulation

We now show how to adapt the weights in Equation 2 using online convex programming. Assuming a standard online learning setup under the logarithmic loss, we define the instantaneous loss given a target $y \in \mathbb{R}$ with respect to a fixed weight vector $w \in \mathbb{R}_+^m$ as

$$\ell(y; w) := -\log \mathrm{PoG}_w(y; f_1(y), \ldots, f_m(y)) \;\equiv\; \log \sigma_{\mathrm{PoG}}^2(w) + \frac{(y - \mu_{\mathrm{PoG}}(w))^2}{\sigma_{\mathrm{PoG}}^2(w)}, \qquad (6)$$

with equivalence following by dropping non-essential constant terms. It is straightforward to show $\ell(y; w)$ is convex in $w$, either directly (as in Appendix B), or by appealing to known properties of the log-partition function for exponential family members [25].

As we are interested in large scale applications, we derive an Online Gradient Descent (OGD) [26] learning scheme to exploit the convexity of the loss in a principled fashion. To apply OGD in our setting, we need to restrict the weights to a choice of compact convex set $\mathcal{W} \subset \mathbb{R}_+^m$. For simplicity of exposition, we focus our presentation on the case where the weight space is defined as

$$\mathcal{W} := \{w \in [0, b]^m \; : \; \|w\|_1 \geq \epsilon\}, \qquad (7)$$

where $0 < \epsilon < 1$ and $b \geq 1$. As $\mathcal{W}$ is formed from the intersection of a scaled hypercube and a half-space, it is a convex set with finite diameter, and is clearly compact and non-empty. OGD works by performing two operations, a gradient step and a projection of the modified weights back into $\mathcal{W}$ if the gradient update pushed them outside of $\mathcal{W}$. This projection is essential, as it is responsible for both ensuring that the weighted product of Gaussians is well-defined (e.g. positive variance) and for providing no-regret guarantees comparable to what was previously achieved for B-GLNs [2].

## 3 G-GLN Neurons

We now introduce a new type of neuron which will constitute the basic learning primitive within a G-GLN. The key idea is that further representational power can be added to a weighted product of Gaussians via a contextual gating procedure. We achieve this by extending the previous weighted product of Gaussians model with an additional type of input, which we call *side information*. The side information will be used by a neuron to select a weight vector to apply for a given example from a table of weight vectors. In typical applications to regression, the side information is defined as the (normalized) input features for an input example: i.e. $z = (x - \bar{x})/\sigma_x$.

More formally, associated with each neuron is a context function $c : \mathcal{Z} \to \mathcal{C}$, where $\mathcal{Z}$ is the set of possible *side information* and $\mathcal{C} = \{0, \ldots, k-1\}$ for some $k \in \mathbb{N}$ is the *context space*. Each neuron $i$ is now parameterized by a weight matrix $W_i = [w_{i,0} \ldots w_{i,k-1}]^\top$ with each row vector $w_{ij} \in \mathcal{W}$ for $0 \leq j < k$. The context function $c$ is responsible for mapping side information $z \in \mathcal{Z}$ to a particular row $w_{i,c(z)}$ of $W_i$, which we then use to weight the Product of Gaussians.

In other words, a G-GLN neuron can be defined in terms of Equation 2 by

$$\mathrm{PoG}_W^c(y; f_1(\cdot), \ldots, f_m(\cdot), z) := \mathrm{PoG}_{w^{c(z)}}(y; f_1(\cdot), \ldots, f_m(\cdot)), \qquad (8)$$

with the associated loss function $-\log(\mathrm{PoG}_W^c(y; f_1(y), \ldots, f_m(y), z))$ inheriting all the properties needed to apply Online Convex Programming directly from Equation 6.

**Half-space Gating.** We restrict our attention to the class of half-space context functions, as in [2]. Given a normal vector $v \in \mathbb{R}^d$ and offset $b \in \mathbb{R}$, consider the associated affine hyperplane $\{z \in \mathbb{R}^d : z \cdot v = b\}$. This divides $\mathbb{R}^d$ in two, giving rise to two half-spaces, one of which we denote $H_{v,b} = \{z \in \mathbb{R}^d : z \cdot v \geq b\}$. The associated half-space context function is then given by $c(z) := 1$ if $z \in H_{v,b}$ or 0 otherwise. Richer notions of context can be created by composition. In particular, any finite set of $s$ context functions $\{c_i : \mathcal{Z} \to \mathcal{C}_i\}_{i=1}^s$ with associated context spaces $\mathcal{C}_1, \ldots, \mathcal{C}_s$ can be composed into a single higher order context function by defining $c(z) = (c_1(z), ..., c_s(z))$. We will refer to the choice of $s$ as the *context dimension*.

**Bias Models.** G-GLN neurons transform an input set of Gaussians to an output Gaussian. Recall that the mean of a product of Gaussian PDFs must lie within the convex hull defined by the means of the individual input Gaussian PDFs (Section 2.1). To ensure the G-GLN neuron can represent any mean in $[-r, r]^D$, where $D$ is the target dimension, we therefore concatenate a number of *bias inputs*,

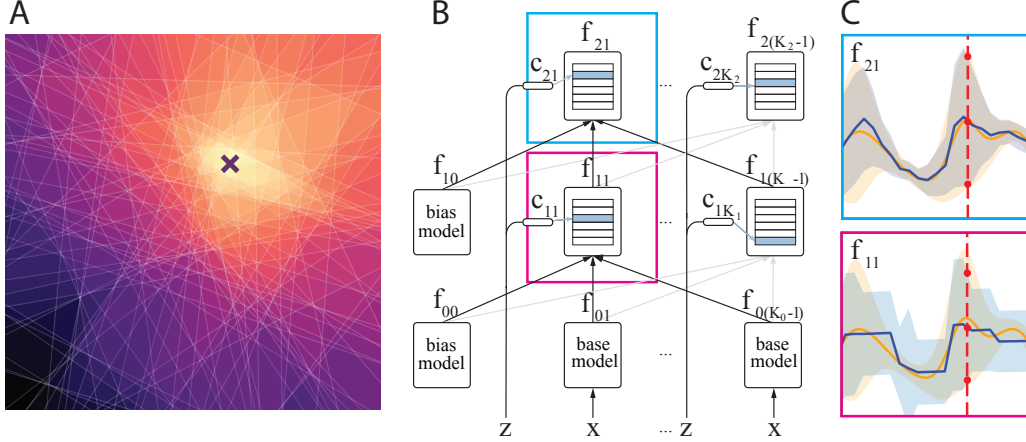

Figure 1: (**A**) Illustration of half-space gating for a 2D context. Color represents how many half-spaces intersect with the data point $x$. Within each region of constant color (each polytope), the gated weights for a G-GLN network are constant. (**B**) G-GLN feed-forward architecture. Each neuron uses its active weights to predict the target density as a function of the preceding layer outputs. (**C**) Illustration of the function sufficient statistics (mean and standard deviation) predicted by two neurons at different G-GLN layers, visualized for both for a single input (red line) and across all inputs within a fixed range (blue). Deeper neurons more accurately reconstruct the true density (orange).

i.e. constant Gaussian PDFs to the input of each neuron. In the univariate case, we concatenate two Gaussian PDFs with mean $\pm r$ with a typical value of $r = 5$ (the target is standardized). This generalizes to the multivariate case by multiplying the two scalars $\pm rD$ against each $D$-dimensional standard basis vector, allowing the convex hull of the bias inputs to span the $[-r, r]^D$ target hypercube.

## 4  G-GLN Architecture

We now describe how the neurons defined in the previous section are assembled to form a G-GLN (Figure 1, B). Similar to its B-GLN predecessor [2, 3], a G-GLN is a feed-forward network of data-dependent distributions. Each neuron calculates the sufficient statistics $(\mu, \sigma^2)$ for its associated PDF using its active weights, given those emitted by neurons in the preceding layer.

**Inputs and Side Information.**    There are two types of input to neurons in the network. The first is the side information, which can be thought of as the input features, and is used to determine the weights used by each neuron via half-space gating. The second is the input to the neuron, which will be the PDFs output by the previous layer, or in the case of layer 0, some provided base models. To apply a G-GLN in a supervised learning setting, we need to map the sequence of input-label pairs $(x_t, y_t)$ for $t = 1, 2, \ldots$ onto a sequence of (side information, base Gaussian PDFs, label) triplets $(z_t, \{f_{0i}\}_i, y_t)$. The side information $z_t$ will be set to the (potentially normalized) input features $x_t$. The Gaussian PDFs for layer 0 will generally include the necessary base Gaussian PDFs to span the target range, and optionally some base prediction PDFs that capture domain-specific knowledge.

**Model Description.**    More formally, a G-GLN consists of $L + 1$ layers indexed by $i \in \{0, \ldots, L\}$, with $K_i$ neurons in each layer. The weight space for a neuron in layer $i$ will be denoted by $\mathcal{W}_i$; the subscript is needed since the dimension of the weight space depends on $K_{i-1}$. Each neuron/distribution will be indexed by its position in the network when laid out on a grid; for example, $f_{ik}$ will refer to the family of PDFs defined by the $k$th neuron in the $i$th layer. Similarly, $c_{ik}$ will refer to the context function associated with each neuron in layers $i \geq 1$, and $\mu_{ik}$ and $\sigma^2_{ik}$ (or $\Sigma_{ik}$ in the multivariate case) referring to the sufficient statistics for each Gaussian PDF.

**Heteroskedastic Regression Example.**    We show an illustrative example on a popular heteroskedastic benchmark function $\mathcal{N}(\mu(x_i), \exp(g(x)))$, with mean $\mu(x) = 2[\exp(-30(x - 0.25)^2) + \sin(\pi x^2)] - 2$ and the logarithm of the standard deviation $g(x) = \sin(2\pi x)$ [27, 28]. Interme-

---

**Algorithm 1** G-GLN: inference with optional update

---

1: **Input:** base model / features $\{\mu_{0j}, \sigma_{0j}\}_{j=0}^{K_0-1}$
2: **Input:** side information $z \in \mathcal{Z}$, target $y \in \mathbb{R}$
3: **Input:** G-GLN weights $\{W_{ik}\}$, learning rate $\eta \in (0, 1)$

4: **Output:** Gaussian PDF

5: **for** $i \in \{1, \ldots, L\}$ **do**
6:      **for** $k \in \{1, \ldots, K_i\}$ **do**
7:          $(w_0, \ldots, w_{K_{i-1}}) \leftarrow W_{ikc_{ik}}(z)$
8:          $\sigma_{ik}^2 \leftarrow \left[ \sum_{j=0}^{K_{i-1}} w_j / \sigma_{i-1,j}^2 \right]^{-1}$
9:          $\mu_{ik} \leftarrow \sigma_{ik}^2 \left[ \sum_{j=0}^{K_{i-1}} w_j \mu_{i-1,j} / \sigma_{i-1,j}^2 \right]$
10:         $W_{ikc_{ik}(z)} \leftarrow \text{PROJ}_i [ W_{ikc_{ik}(z)} - \eta \nabla \ell_{ik}(y; z)]$ // (if learning)
11:      **end for**
12: **end for**

13: **return** $\mathcal{N}(\mu_{L1}, \sigma_{L1}^2)$

---

diate layer outputs in the G-GLN are illustrated in Figure 1(C). For each training input $x$ (red line), with target $y$ (intersection of dashed red line and yellow curve), and for each neuron: (1) a set of active weights are selected by applying the context function to the broadcast side information (in this case simply $x$), (2) the active weights are used to predict the target distribution as a function of preceding predictions, and (3) the active weights are updated with respect to the loss function defined in Equation (6). Figure 1(B) compares the predictions (blue) for all values of $x$ for two individual neurons. It is clearly evident from inspection that neurons in higher layers produce more accurate predictions of the sufficient statistics given only the preceding predictions as input.

**Generating Context Functions.** We sample our context functions randomly according to the scheme first introduced in [2, 3], which is inspired by the SimHash method [29] for locality sensitive hashing. Recall that a half-space context is defined by $H_{v,b}$; to sample $v$, we first generate an i.i.d. random vector $x = (x_1, ..., x_d)$ of dimension $d$, with each component of $x$ distributed according to the unit normal $\mathcal{N}(0, 1)$, and then divide by its 2-norm, giving us a vector $v = x / ||x||_2$. This scheme uniformly samples points from the surface of a unit sphere. The scalar $b$ is sampled directly from a standard normal distribution.

To gain intuition for this procedure, consider Figure 1(A). There is a 1-1 mapping between any convex polytope formed from the intersection of each of the half-spaces, and the collective firing pattern of all context functions in the network. Choices of side information close in terms of cosine similarity will map to similar sets of weights. A (local) update of the weights corresponding to a particular convex region will therefore affect neighbouring regions, but with decreasing impact in proportion to the number of overlapping half-spaces.

## 5 G-GLN Algorithm

We now describe how inference is performed in a G-GLN. For layer 0, we assume all the base models are given. For layers $i \geq 1$, we then have

$$f_{ik}(y; z) := \text{PoG}_{W_{ik}}^{c_{ik}} (y; f_{i-1,0}(\cdot; z), \ldots, f_{i-1,K-1}(\cdot; z), z). \tag{9}$$

Equation 9 makes it explicit that, conceptually, a G-GLN is a network of Gaussian PDFs, each of which depend on the side information $z$ via gating. Computationally, this involves a forward pass of the network to compute the relevant sufficient statistics for each neuron (using Equations 3-5). By re-expressing Equation 9 as

$$f_{ik}(y; z) \propto \exp \left\{ \log \prod_{j=1}^{K_{i-1}} [f_{i-1,j}(y; z)]^{W_{ijc_{ik}(z)}} \right\} = \exp \left\{ \sum_{j=1}^{K_{i-1}} W_{ijc_{ik}(z)} \log (f_{i-1,j}(y; z)) \right\},$$

one can view each neuron as having an exponential output non-linearity and a logarithmic input non-linearity. Since these non-linearities are inverses of each other, stacking layers causes the non-

Table 1: Test RMSE and standard errors for G-GLN versus three previously published methods on a standard suite of UCI regression benchmarks with $N$ instances of $d$ features each. Models are trained for 40 epochs and results summarized for 20 random seeds (5 for Protein).

| Dataset | $N$ | $d$ | G-GLN | VI [30] | PBP [31] | DO [32] |
|---|---|---|---|---|---|---|
| Boston Housing | 506 | 13 | **2.84**±0.03 | 4.32±0.29 | 3.01±0.18 | 2.97±0.19 |
| Concrete Compression Strength | 1030 | 8 | 5.84±0.03 | 7.13±0.12 | 5.67±0.09 | **5.23**±0.12 |
| Energy Effiency | 768 | 8 | **1.31**±0.01 | 2.65±0.08 | 1.80±0.05 | 1.66±0.04 |
| Kin8nm | 8192 | 8 | **0.09**±0.00 | 0.10±0.00 | 0.10±0.00 | 0.10±0.00 |
| Naval Propulsion | 11,934 | 16 | **0.00**±0.00 | 0.01±0.00 | 0.01±0.00 | 0.01±0.00 |
| Combined Cycle Power Plant | 9568 | 4 | **3.90**±0.01 | 4.33±0.04 | 4.12±0.03 | 4.02±0.04 |
| Protein Structure | 45,730 | 9 | **3.77**±0.01 | 4.84±0.03 | 4.73±0.01 | 4.36±0.01 |
| Wine Quality Red | 1599 | 11 | **0.57**±0.00 | 0.65±0.01 | 0.64±0.01 | 0.62±0.01 |
| Yacht Hydrodynamics | 308 | 6 | 3.76±0.04 | 6.89±0.67 | **1.01**±0.05 | 1.11±0.09 |

linearities to cancel, so the density output by a G-GLN collapses to a linear function of the gated weights (i.e. a Gated Linear Network). The same cancellation argument applies to B-GLN [2], where the output and input non-linearities are the sigmoid and logit functions.

A distinguishing feature of a G-GLN is that every neuron directly attempts to predict the target, by locally boosting the accuracy of its input distributions. Because of this, every neuron will have its own loss function defined only in terms of its own weights. Given a (potentially vector-valued) target $y$, and side information $z$ (which will typically be identified with the input features), each neuron-specific loss function will be

$$\ell_{ik}(y;z) := -\log f_{ik}(y\,;\,z). \tag{10}$$

This loss can be optimized using online gradient descent [26], which involves performing a step of gradient descent, and projecting the weights back onto $\mathcal{W}_i$, via the update rule

$$W_{ikc_{ik}(z)} \leftarrow \text{PROJ}_i[W_{ikc_{ik}(z)} - \eta \nabla \ell_{ik}(y;z)], \tag{11}$$

where $W_{ikj}$ refers to the $j$th row of the neurons weight matrix $W_{ik}$, $\eta > 0$ is the learning rate and $\text{PROJ}_i[w] := \arg\min_{w' \in \mathcal{W}_i} \|w' - w\|_2$ is the projection operator with respect to the Euclidean norm.

Algorithm 1 provides pseuodocode for both inference and (optionally) weight adaptation for a univariate G-GLN for a given input, with the top-most neuron taken as the final Gaussian PDF. The multivariate case can be obtained by replacing lines 8-9 with Equation 4 or 5. The total time complexity to perform inference is the sum of the cost of computing the gating operations $O\left(d\left(\sum_{i=1}^{L} K_i\right)\right)$, where $d$ is the dimensionality of the input vector, and the cost of propagating the sufficient statistics through the network, $O\left(\sum_{i=1}^{L} K_i K_{i-1}\right)$.

## 6 Experimental Results

We applied G-GLNs to univariate regression, multivariate regression, contextual bandits with real valued rewards, denoising and image infilling. Model, experimental and implementation details common across our test domains are discussed below.

**Training Setup.** Weights for all neurons in layer $i$ are initialized to $1/K_{i-1}$ where $K_{i-1}$ is the number of neurons in the previous layer. Note that due to the convexity of the loss, the choice of initial weights plays a less prominent role in terms of overall performance compared with typical deep learning applications. The only source of non-determinism in the model is the choice of context function; to address this, all of our results are reported by averaging over multiple random seeds. For regression experiments, multiple epochs of training are used. Training data is randomly shuffled at the beginning of each epoch, and each example is seen exactly once within an epoch.

**Bias and Base Predictions.** Constant bias inputs for each neuron were set to span the target range, as described in Section 3. Given $d$-dimensional input of the form $(x_1, x_2, \ldots, x_d)$, we adopted the

Table 2: (**Left**) Test MSE for G-GLN versus previously published methods on the SARCOS inverse dynamics dataset [35]. G-GLNs are trained for 2000 epochs using the same test procedure as [36]. (**Right**) Performance of a G-GLN based GLCB algorithm for the continuous contextual bandits tasks and competitors described in [1, 37]. Ranks are computed by running each algorithm on 500 randomly sampled environments. Raw scores are provided in Table 3 of the Appendix.

| Algorithm | MSE | Algorithm | financial | jester | wheel | mean rank |
|---|---|---|---|---|---|---|
| **G-GLN** | **0.10** | **G-GLN** | 3 | **1** | 2 | **2** |
| Random forest | 2.39 | BBAlphaDiv | 10 | 9 | 10 | 9.67 |
| MLP | 2.13 | constSGD | 9 | 8 | 6 | 7.67 |
| Stochastic decision tree | 2.11 | ParamNoise | 7 | 10 | 4 | 7 |
| Gradient boosted tree | 1.44 | BBB | 8 | 5 | 6 | 6.33 |
| TabNet-S | 1.25 | NeuralGreedy | 5 | 4 | 9 | 6 |
| Adaptive neural tree | 1.23 | BootRMS | 4 | 2 | 8 | 4.67 |
| TabNet-M | 0.28 | Dropout | 6 | 3 | 5 | 4.67 |
| TabNet-L | 0.14 | NeuralLinear | 2 | 7 | 3 | 4 |
| | | LinFullPost | **1** | 6 | **1** | 2.67 |

convention of adding $d$ Gaussian PDF *base predictions* to layer 0. The mean and variance of the $j$th expert was calculated online either by setting each expert to be centered at a single input feature $x_j$ and with a fixed width $\sigma$ such as 1.0, or from an analytic formula that applies Bayesian Linear Regression (BLR) to learn a mapping of $x_j$ to an approximation of the target distribution.

**Output Aggregation and Weight Projection.** As each neuron in a G-GLN models the target distribution, any choice of neuron to be the output provides an estimate of the target density; we either take the output of the top-most neuron or use the switching aggregation method introduced in [2] for B-GLNs which uses Bayesian tracking [33] to estimate the best performing neuron on recent data. See Appendix D for details of switching aggregation.

We explored multiple methods for implementing weight projection efficiently, and obtained the best performance in our regression benchmarks by an approximate solution which used the log-barrier method [34]. This method essentially amounts to adding an additional regularization term to the loss, which has negligible affect on the cost of inference; see Appendix E for implementation details.

## 6.1 UCI Regression

We begin by evaluating the performance of a G-GLN to solve a benchmark suite of univariate UCI regression tasks. We adopt the same datasets and training setup described in [32], and compare G-GLN performance to the previously published results for 3 MLP-based probabilistic methods: variational inference (VI) [30], probabilistic backpropagation (PBP) [31] and the interpretation of dropout (DO) as Bayesian approximation as described in [32]. Our results are presented in Table 1. It is evident that G-GLN achieves competitive performance, outperforming PBP, BP and DO on 7 out of 9 regression tasks. See Appendix H.1 for full details.

## 6.2 Inverse Dynamics

Next we demonstrate G-GLNs on regression tasks where both the inputs and targets are multi-dimensional. We consider the SARCOS dataset for a 7 degree-of-freedom robotic arm [35]: using a 21-dimensional feature vector (7 joint positions, velocities and accelerations) to predict the 7 joint torques. We compare our performance to the state-of-the-art TabNet model [36] and the same suite of standard regression algorithms considered by the TabNet authors. See Appendix H.2 for details.

Table 2 (left) shows that G-GLN outperforms all of the baselines, including the largest TabNet, which is a complex system of neural networks optimized for tabular data, exploiting residual transformer blocks for sequential attention. It is likely that a similar system could exploit G-GLNs as components for improved performance, but doing so is beyond the scope of this paper.

## 6.3 Online Contextual Bandits

The authors of [1] proposed an algorithm, Gated Linear Context Bandits (GLCB), by which B-GLNs could be applied to solve contextual bandits tasks with binary rewards. GLCB provides a

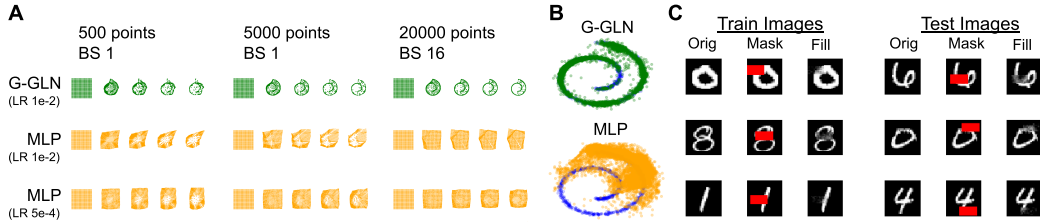

Figure 2: Denoising multi-dimensional data with G-GLNs. (**A**) G-GLNs (top row) and MLPs (bottom two rows) are trained on 1-step denoising of a Swiss Roll density under additive Gaussian noise (BS = batch size, LR = learning rate). Starting with a grid, the original Swiss Roll data manifold is reconstructed with multi-step denoising. Larger version in Appendix G.2. (**B**) Sampling via HMC using the gradient field inferred by denoising. Shown are samples from G-GLN inferred gradient (green), MLP inferred gradient (orange) and original data manifold (blue). (**C**) Infilling of MNIST train images (left) or unseen test images (right) is shown for binary occlusion masks, after training a G-GLN for only one epoch over the dataset with batch size 1 to remove additive Gaussian noise from each train image. Orig: original image. Mask: masked: Fill: filled. More examples in Appendix G.3

UCB-like [38] rule that exploits GLN half-space activation as a "pseudo-count" that is shown to be effective for exploration (full details in Appendix G.1). Our G-GLN provides a natural solution for extending GLCB to continuous rewards. Table 2 (right) compares the results of a G-GLN based GLCB algorithm (see Appendix H.3 for details) to three bandits tasks derived from UCI regression datasets, a standard benchmark in previous literature. G-GLN obtains the best mean rank across these tasks compared to 9 popular Bayesian deep learning methods [37]. Similar to [1], our results are obtained in an online regime – each data point is considered once without storage, whereas all other methods were able to i.i.d. resample from prior experience to learn an effective representation.

### 6.4 Application to Denoising Density Estimation

One application of high-dimensional regression is to the problem of density estimation via denoising [39, 40, 41, 42], which gives the ability to sample any conditional distribution from a learnt gradient of the log-joint data distribution. We use G-GLNs to approximate this score function, $\nabla_x \log p(x)$ [43], by using a G-GLN multivariate regression model as a denoising autoencoder [39, 40, 41, 42]. We train the GLN by adding isotropic Gaussian noise with covariance $\lambda \mathcal{I}$ ($0 < \lambda \ll 1$) to each data point and regressing to the un-noised point. At convergence, the vector $(x - \mu_{L,1}(x))/\lambda$ approximates the score function [44], which we can feed into Hamiltonian Monte Carlo (HMC) [45] to approximately sample from the distribution implied by the score field. See Appendix F for details.

From Figure 2(A) it is evident that G-GLNs can learn reasonable approximate gradient fields for 2D distributions from just a single online pass of 500-5000 samples. Starting from a grid, multi-step denoising can then by applied to reconstruct the original data manifold. MLPs trained with the same data required a larger batch size and many more samples to accurately approximate the data density. This is evident in Figure 2(B), which shows the result of HMC sampling [45] using the G-GLN versus MLP estimated gradient fields. Figure 2(C) demonstrates that the same process can be extended to much higher-dimensional problems, e.g. MNIST density modelling: iterative G-GLN denoising can be leveraged to fill in occluded regions in MNIST train or unseen test images after a single online pass through the train set in which it is trained to remove small additive Gaussian noise patterns from each image. This suggests an exciting avenue for future work applying G-GLNs as data-efficient pattern completion memories.

## 7 Conclusion

We have introduced a new backpropagation-free deep learning algorithm for multivariate regression that leverages local convex optimization and data-dependent gating to model highly non-linear and heteroskedastic functions. We demonstrate competitive or state-of-the-art performance on a comprehensive suite of established benchmarks. The simplicity and data efficiency of the G-GLN approach, coupled with its strong performance in high-dimensional multivariate settings, makes us optimistic about future extensions to a broad range of applications.

## Software

All models implemented using JAX [46] and the DeepMind JAX Ecosystem [47, 48, 49, 50]. Open source GLN implementations (including G-GLN) are available at: `www.github.com/deepmind/deepmind-research`.

## Broader Impact

Regression models have long been ubiquitous in both industry and academia, and we are optimistic that our work can provide improvement to existing practice and results. Like any supervised learning technique, the output of this model is a function of its input data, so appropriate due diligence is required during all stages of data collection, training and deployment, e.g. with respect to issues of algorithmic fairness and bias, as well as safety and robustness.

## Acknowledgments

We thank Agnieszka Grabska-Barwinska, Chris Mattern, Jianan Wang, Pedro Ortega and Marcus Hutter for helpful discussions.

## Funding Disclosure

All authors are employees of DeepMind.

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
