[Supplementary Material]

## A    Weighted Products of Gaussians

A well-known result is that a product of Gaussian PDFs collapses to a scaled Gaussian PDF (e.g. [24]). In particular, if we define

$$\sigma^2_{\text{PoG}} := \left[\sum_{i=1}^{m} \frac{1}{\sigma_i^2}\right]^{-1} \quad \text{and} \quad \mu_{\text{PoG}} := \sigma^2_{\text{PoG}}\left[\sum_{i=1}^{m} \frac{\mu_i}{\sigma_i^2}\right], \tag{12}$$

and let $f_{\text{PoG}}(\cdot)$ denote the associated PDF of $\mathcal{N}(\mu_{\text{PoG}}, \sigma_{\text{PoG}})$, then we have that $f_{\text{PoG}}(y) \propto \prod_{i=1}^{m} f_i(y)$. In the case where $w = \vec{1}$, this implies that $\text{PoG}_{\vec{1}}(y\,;\,\dots) = f_{\text{PoG}}(y)$ as the constant of proportionality (not a function of $y$) is cancelled out by the division by Z in Equation 2, and we are left with an integral of a PDF in the denominator. Now consider a Gaussian PDF $f$ raised to a power $p \in \mathbb{R}_+$, i.e.

$$f^p(y) = \left[\frac{1}{\sigma\sqrt{2\pi}}\exp\left\{-\frac{1}{2}\left(\frac{y-\mu}{\sigma}\right)^2\right\}\right]^p \propto \exp\left\{-\frac{1}{2}\frac{(y-\mu)^2}{\sigma^2\,p^{-1}}\right\},$$

which corresponds to an unnormalized Gaussian PDF with mean $\mu$ and variance $\sigma^2\,p^{-1}$. Thus we can replace each $f_i(y)^{w_i}$ term in Equation 2 with the PDF associated with $\mathcal{N}(\mu, \sigma^2\,p^{-1})$. Combining the above techniques for products and powers allows us to exactly interpret the weighted product of experts as another Gaussian expert $\mathcal{N}\left(\mu_{\text{PoG}}(w), \sigma^2_{\text{PoG}}(w)\right)$ where

$$\sigma^2_{\text{PoG}}(w) := \left[\sum_{i=1}^{m} \frac{w_i}{\sigma_i^2}\right]^{-1} \quad \text{and} \quad \mu_{\text{PoG}}(w) := \sigma^2_{\text{PoG}}(w)\left[\sum_{i=1}^{m} \frac{w_i\,\mu_i}{\sigma_i^2}\right]. \tag{13}$$

## B    Properties of the G-GLN Loss

**Gradient.**    First define $\omega_i := w_i/\sigma_i^2$, $\omega = (\omega_1, \dots, \omega_m)$ and $\mu = (\mu_1, \dots, \mu_m)$, which due to the non-negativity of $w_i$ implies $\|\omega\|_1 = \sum_i \omega_i$. Hence $\sigma^2_{\text{PoG}} = \|\omega\|_1^{-1}$ and $\mu_{\text{PoG}} = \omega^T\mu\,/\,\|\omega\|_1$. Using this notation, we can reformulate Equation 6 as

$$\ell(y; \omega) = -\log\|\omega\|_1 + \left(x - \omega^T\mu/\,\|\omega\|_1\right)^2\|\omega\|_1. \tag{14}$$

The first partial derivative can be obtained by direct calculation, and is

$$\frac{\partial\ell(y;\cdot)}{\partial\omega_i} = -\|\omega\|_1^{-1} + \left(y - \omega^T\mu/\,\|\omega\|_1\right)\left(y - 2\mu_i + \omega^T\mu/\,\|\omega\|_1\right).$$

Hence, using the above and $\frac{\partial\ell(y;w)}{\partial w_i} = \frac{\partial\ell(y;\omega)}{\partial\omega_i}\frac{\partial\omega_i}{\partial w_i} = \frac{\partial\ell(y;\omega)}{\partial\omega_i}\frac{1}{\sigma_i^2}$, we have

$$\nabla_w\,\ell(y; w) = \text{diag}\left(\frac{1}{\sigma^2}\right)\left[(y - \mu_{\text{PoG}})(\mathbb{1}_{m,1}(y + \mu_{\text{PoG}}) - 2\mu) - \mathbb{1}_{m,1}\,\sigma^2_{\text{PoG}}\right].$$

**Convexity.**    Here we prove that $\ell(y; w)$ is a convex function of $w$ by showing that the Hessian of Equation 14 is positive semi-definite (PSD). Let $g(\omega) := \omega^T\mu\,/\,\|\omega\|_1$ and $g'(\omega) := \frac{\partial g}{\partial\omega_j} = \mu_j\|\omega\|_1^{-1} - \omega^T\mu\|\omega\|_1^{-2}$, which allows us to compute the second partial derivative as

$$\frac{\partial^2\ell(y;\cdot)}{\partial\omega_i\partial\omega_j} = \|\omega\|_1^{-2} - g'(\omega)y + 2g'(\omega)\mu_i + g'(\omega)y - 2g(\omega)g'(\omega)$$

$$= \|\omega\|_1^{-2} + 2g'(\omega)(\mu_i - g(\omega))$$

$$= \|\omega\|_1^{-2} + 2(\mu_j\|\omega\|_1^{-1} - \omega^T\mu\|\omega\|_1^{-2})(\mu_i - \omega^T\mu/\,\|\omega\|_1)$$

$$= \|\omega\|_1^{-2} + 2\|\omega\|_1^{-1}(\mu_j - \omega^T\mu/\,\|\omega\|_1)(\mu_i - \omega^T\mu/\,\|\omega\|_1).$$

Thus the Hessian of Equation 14 is

$$\nabla^2\ell(y;\omega) = \|\omega\|_1^{-2}\mathbb{1}_{m,m} + 2\|\omega\|_1^{-1}(\mu - \omega^T\mu/\,\|\omega\|_1\,\mathbb{1}_{m,1})(\mu - \omega^T\mu/\,\|\omega\|_1\,\mathbb{1}_{m,1})^T, \tag{15}$$

where $\mathbb{1}_{m,n}$ denotes the $m \times n$ matrix whose entries are all 1. As $\mathbb{1}_{m,m}$ is PSD and $\|\omega\|_1^{-2} > 0$, the first additive term is PSD. The second term is also PSD, since $2\|\omega\|_1^{-1} > 0$ and the outer product $(\mu - \omega^T\mu/\,\|\omega\|_1\,\mathbb{1}_{m,1})(\mu - \omega^T\mu/\,\|\omega\|_1\,\mathbb{1}_{m,1})^T$ is PSD by letting $a = (\mu - \omega^T\mu/\,\|\omega\|_1\,\mathbb{1}_{m,1})$ and observing that $u^\top a a^\top u = (u^\top a)(u^\top a)^\top = (u \cdot a)^2 \geq 0$ for all $u \in \mathbb{R}^m$. Hence since the Hessian is the sum of two PSD matrices, it is PSD which implies that $\ell(y;\omega)$ and therefore $\ell(y;w)$ is a convex function of $w$.

## C   Learning the Base Model

Every neuron in a G-GLN takes one-or-more Gaussian PDFs as input and produces a Gaussian PDF as output. This raises the question of what input to provide to neurons in the first layer, i.e. the *base prediction*. We consider three solutions: (1) None. The input sufficient statistics to each neuron are already concatenated with so-called "bias" Gaussians to ensure that the target mean falls within the convex hull defined by the input means (described in Section 3). (2) A Gaussian PDF for each component $x_i$ of the input vector, with $\mu = x_i$ and $\sigma = $ constant. It is perhaps surprising that the neuron inputs are not required to be a function of the $x_i$s, but this is permissible because $x_i$ is z-score normalized and broadcast to every neuron as side information $z_i$.

We present a third option (3) whereby the base prediction is provided by a probabilistic *base model* trained to directly predict the target using only a single feature dimensions. The formulation of this Bayesian Linear-Gaussian Regression (BLR) model is described below. Empirically we find that it leads to improved data efficiency in the first epoch of training (see examples in Figure 3) with only an additional $\mathcal{O}(1)$ time and space cost per feature dimension.

Consider a dataset $\mathcal{D} = \{x_i, y_i\}_{i=1}^N$ of zero-centered univariate features $x_i \in \mathbb{R}$ and corresponding targets $y_i \in \mathbb{R}$. We assume a Normal-linear relationship between a feature $x_i$ and target $y_i$,

$$y_i \sim \mathcal{N}(\theta x_i + \beta, \tau^{-1})$$

where $\theta$ and $\beta$ are some coefficients, and $\tau$ is the precision (inverse variance). We assume $\tau$ is known, but it can also be optimized via (type II) maximum likelihood estimation. We also assume an isotropic Normal prior over $\theta$ and $\beta$, i.e. $\theta \sim \mathcal{N}(0, \tau_0^{-1})$ and $b \sim \mathcal{N}(0, \tau_0^{-1})$, where $\tau_0$ is the prior precision.

By adapting widely known equations (e.g. Equations 3.53-3.54 in [51]) we can obtain the posterior for $\theta$ as

$$p(\theta \mid \mathcal{D}) = \mathcal{N}(\theta \mid \mu_\theta, \tau_\theta^{-1})$$
$$\mu_\theta = \tau \tau_\theta^{-1} \sum_{x_i, y_i \in \mathcal{D}} x_i y_i$$
$$\tau_\theta = \tau_0 + \tau \sum_{x_i \in \mathcal{D}} x_i^2 \ .$$

Similarly, we obtain the posterior for $\beta$ as

$$p(\beta \mid \mathcal{D}) = \mathcal{N}(\beta \mid \mu_\beta, \tau_\beta^{-1})$$
$$\mu_\beta = \tau \tau_\beta^{-1} \sum_{y_i \in \mathcal{D}} y_i$$
$$\tau_\beta = \tau_0 + \tau N \ .$$

Putting these two together, we can obtain the posterior predictive distribution,

$$p(y \mid x, \mathcal{D}) = \mathcal{N}(y \mid \mu_\theta x + \mu_\beta, x^2 \tau_\theta^{-1} + \tau_\beta^{-1} + \tau^{-1}) \ .$$

It is apparent that updates and inference can be performed incrementally in constant time and space by storing and updating the sufficient statistics $\sum_i x_i y_i$, $\sum_i x_i^2$, $\sum_i y_i$, $\sum_i 1$.

We can use this BLR formulation to convert the input features into probability densities. Specifically, for each feature, we independently maintain posterior/sufficient statistics and use the posterior predictive distributions as inputs to the base layer of the G-GLN.

## D   Switching Aggregation

Because every neuron in a G-GLN directly models the target distribution, there is no one natural definition of the network output. One convention is simply to have a final layer consisting of a single neuron, and take the output of that neuron as the network output. An alternative method of *switching aggregation* was used in [2, 52], whereby an incremental online update rule was used to weight the contributions of individual neurons in the network to an overall estimate of the target density.

Figure 3: Effect of using Bayesian linear regression (BLR) versus a constant base model $\mathcal{N}(0, 1)$ on predictive RMSE for four UCI regression tasks. Results are shown for the first epoch of training.

We extend the switching aggregation procedure from the Bernoulli to Gaussian case by replacing a Bernoulli target probability value with a Gaussian probability density value evaluated at the target. The switching algorithm of [52] was originally presented in terms of log-marginal probabilities, which can cause numerical difficulties at implementation time. Instead we use an equivalent formulation derived from [2] that incrementally maintains a weight vector that is used to compute a convex combination of model predictions, i.e. the densities given by each neuron in the network, at each time step.

Using notation similar to [2], let $m \geq 2$ denote the number of neurons, and $w_t^i \in [0, 1]$ denote the weight associated with model $i$ at times $t \geq 1$. The density output by the $i$th neuron at time $t$, evaluated on target $y_t$, will be denoted by $\rho_i(y_t \mid y_{<t})$. At each time step $t$, switching aggregation outputs the density

$$\pi(y_t \mid y_{<t}) := \sum_{i=1}^{m} w_t^i \, \rho(y_t \mid y_{<t}),$$

with the weights defined, for all $1 \leq i \leq m$, by $w_1^i := 1/m$ and

$$w_{t+1}^i = \frac{\alpha_{t+1}}{m-1} + \left( (1 - \alpha_{t+1}) - \frac{\alpha_{t+1}}{m-1} \right) \frac{w_t^i \, \rho_i(y_t|y_{<t})}{\pi(y_t \mid y_{<t})},$$

with $\alpha_t := 1/t$. This can be implemented in linear time with respect to the number of neurons. Notice that mathematically the weights satisfy the invariant $\sum_{i=1}^{m} w_t^i = 1$ for all times $t \geq 1$, which should be explicitly enforced after each update to avoid numerical issues in any practical implementation.

## E    Weight Projection

Weight projection after an update (Line 11 in Algorithm 1) enforces three sets of constraints: each weight to be in $[0, b]$, mixed means $\mu_{\text{PoG}}$ to be in $[\mu_{\min}, \mu_{\max}]$, and mixed variances $\sigma_{\text{PoG}}^2$ to be in $[\sigma_{\min}^2, \sigma_{\max}^2]$. These constraints ensure that the online convex optimization is well-behaved by forming a convex feasible set and also preventing numerical issues that arise from rounding likelihoods $\mathcal{N}(x; \mu_{\text{PoG}}, \sigma_{\text{PoG}})$ to 0. We outline two ways in which these constraints can be implemented below.

The constraints can be represented in terms of linear inequalities $Aw \leq u$, where $w = W_{ijc_{ij}(z)}$ is the weight vector of neuron $\langle i, j \rangle$ given side info $z$. Assume $w$ violates some of the constraints, therefore we would like to project $w$ onto our feasible set $\{w' : Aw' \leq u\}$. Let $A'$ and $u'$ be the matrix/vector composed of rows/elements of $A$ and $u$ respectively that violate our original inequality, thus $A'w > u'$. Then we can write down the projection problem as $\arg\min_{w'} ||w' - w||^2$ s.t. $A'w' = u'$, the solution of which is $w - A^\dagger(A'w - u')$ where $A^\dagger = A'^T(A'A'^T)^{-1}$ is the pseudo-inverse of $A'$. This pseudo-inverse can be computed efficiently, because all but (at most) two rows of $A'$ are "one-hot".

The exact projection approach relies on dynamically shaped $A'$ and $u'$, support for which is limited in contemporary differentiable programming libraries such as Tensorflow [53] and JAX [46]. Therefore, we take an alternative approach and enforce the inequalities via using logarithmic barrier functions (log-barriers) that augment the original loss function by penalizing the weights that are close to the constraints. Let $A_k$ and $u_k$ be the $k$th row and element of $A$ and $u$ respectively. For the constraint

$A_k^T w \le u_k$, we can define a barrier function

$$\phi_k(w) = \begin{cases} -\log(u_k - A_k^T w) & A_k^T w < u_k \\ +\infty & \text{otherwise} \end{cases}.$$

Note that we are now dealing with strict inequalities rather than $\le$ for convenience. We can then augment the loss function $\ell(y; w)$ from Equation 6, incorporating the barriers,

$$\ell_{\text{combined}}(y; w) = \ell(y; w) + \xi \Phi(w) \tag{16}$$

where $\Phi(w) = \sum_k \phi_k(w)$ and $\xi > 0$ is the barrier constant. Note that $\ell_{\text{combined}}(y; w)$ is convex in $w$ as each $\phi_k(w)$ is convex.

The weight updates can be carried out via $w \leftarrow w - \eta \nabla \ell_{\text{combined}}(y; w)$. For sufficiently small $\eta$ and sufficiently large $\xi$, we will not need the projection step in Line 11 of Algorithm 1, as the constraints are incorporated into the loss function. However, in practice, we need backstops in case weights pass through the barriers due to large gradient steps. We implement the backstops by first hard-clipping each weight to be in $[0, b]$ then by enforcing $\sigma_{\min}^{-2} > \sigma_{\text{PoG}}^{-2} = w^T \sigma_i^{-2} > \sigma_{\max}^{-2}$, which corresponds to performing a single linear projection if the inequality is violated.

## F   Denoising Density Estimation

With $\hat{p}$ denoting a Gaussian likelihood function (as parameterized by a G-GLN) and $p^d(x)$ an unknown data-generating distribution, suppose we add isotropic Gaussian noise of variance $\lambda$ to sampled data points and then denoise them back to the original samples. The expected loss is

$$\mathbb{E}_{x \sim p^d(x)} \left[ \mathbb{E}_{\xi \sim \mathcal{N}(0,\lambda)} \left[ \ln \hat{p}(x \mid z, x + \xi) \right] \right]$$

$$= \mathbb{E}_{x \sim p^d(x)} \left[ \mathbb{E}_{\xi \sim \mathcal{N}(0,\lambda)} \left[ \ln \frac{\exp(-\|x - \mu(x + \xi)\|^2 / (2\sigma^2))}{(2\pi\sigma^2)^{(d/2)}} \right] \right]$$

$$= \mathbb{E}_{x \sim p^d(x)} \left[ \mathbb{E}_{\xi \sim \mathcal{N}(0,\lambda)} \left[ -\|x - \mu(x + \xi)\|^2 / (2\sigma^2(x + \xi)) - (d/2)\ln(2\pi\sigma^2(x + \xi)) \right] \right].$$

Taking the variational derivative of this expected loss with respect to our G-GLN demonstrates the relationship between the value of the optimal output $\mu(x)$ and the gradient of the log data density:

$$0 = \mathbb{E}_\xi \left[ p^d(x - \xi)(x - \xi - \mu(x)) \right]$$

$$= \mathbb{E}_\xi \left[ (p^d(x) - \nabla_x p^d(x) \cdot \xi + \mathcal{O}(\|\xi\|^2))(x - \xi - \mu(x)) \right]$$

$$\implies \mu(x) = \frac{p^d(x)x + \nabla_x p^d(x)\lambda}{p^d(x)}$$

$$= x + \lambda \nabla_x \ln p^d(x), \tag{17}$$

in the limit $\|\xi\|_2 \to 0$. Therefore, we can approximate the gradient field as $(\mu(x) - x)/\lambda$, which we use in the main text. Hamiltonian Monte Carlo sampling then takes as input this gradient estimate for $\nabla_x \ln p^d(x)$. Denoising iteratively applies the G-GLN, trained on denoising, to an arbitrary starting point $x \to \mu(x) \to \mu(\mu(x))$, and so on.

## G   Additional Results

### G.1   Contextual Bandits

In [1] the authors present a B-GLN based algorithm, GLCB, that achieves state-of-the-art results across a suite of contextual bandits tasks with both binary and real-valued rewards. The former uses the B-GLN formulation directly. For the latter, the authors present an algorithm called CTree for tree-based discretization, i.e. using $b - 1$ B-GLNS arranged within a binary tree structure to model the target distribution over $b$ bins. In both cases, GLCB leveraged properties of GLN half-space gating to derive a UCB-like [38] rule based on "pseudo-counts" (inspired by [54]) to help guide exploration. At each timestep $t$, the GLCB policy [1] greedily maximizes a linear combination of the expected action reward as predicted by a GLN and an exploration bonus $\sqrt{\log t / \hat{N}(s_t, a)}$ where $\hat{N}(s_t, a)$ is the pseudocount term capturing how similar the current context-action pair $\langle s_t, a \rangle$ is to the previously seen data. This term is computed at no additional cost by utilizing gating functions of GLN neurons.

Table 3: Performance of the GLN-based GLCB algorithms for the contextual bandits tasks and competitors described in [1, 37]. G-GLCB uses a single G-GLN instead of a CTree of 7 equivalent-sized B-GLNs (italics), the method described in [1], to model continuous-valued results. Results are mean and standard error of cumulative rewards over 500 random environment seeds.

|  | Binary targets | | | | Continuous targets | | |
| --- | --- | --- | --- | --- | --- | --- | --- |
| **Algorithm** | adult | census | covertype | statlog | financial | jester | wheel |
| G-GLN | - | - | - | - | 3018±3 | **3301±4** | 4386±11 |
| B-GLN | **678±5** | **2718±3** | 2715±12 | **4863±1** | *3038±3* | *3298±3* | *4432±11* |
| BBAlphaDiv | 18±2 | 932±12 | 1838±9 | 2731±15 | 1860±1 | 3112±4 | 1776±11 |
| BBB | 399±8 | 2258±12 | 2983±11 | 4576±10 | 2172±18 | 3199±4 | 2265±44 |
| BootRMS | 676±3 | 2693±3 | **3002±7** | 4583±11 | 2898±4 | 3269±4 | 1933±44 |
| Dropout | 652±5 | 2644±8 | 2899±7 | 4403±15 | 2769±4 | 3268±4 | 2383±48 |
| LinFullPost | 463±2 | 1898±2 | 2821±6 | 4457±2 | **3122±1** | 3193±4 | **4491±15** |
| NeuralGreedy | 598±5 | 2604±14 | 2923±8 | 4392±17 | 2857±5 | 3266±8 | 1863±44 |
| NeuralLinear | 391±2 | 2418±2 | 2791±6 | 4762±2 | 3059±2 | 3169±4 | 4285±18 |
| ParamNoise | 273±3 | 2284±5 | 2493±5 | 4098±10 | 2224±2 | 3084±4 | 3443±20 |
| constSGD | 107±3 | 1399±22 | 1991±9 | 3896±18 | 1862±1 | 3136±4 | 2265±31 |

Table 3 expands on the results in Section 6.3 to demonstrate the performance of GLNs for both binary and continuous-valued rewards. It is evident that GLNs achieve state-of-the-art performance in both regimes. Moreover, using the natural G-GLN formulation described in this paper is able to match the previous performance of a CTree of B-GLNs with just a single equivalent-sized network (an order-of-magnitude reduction in memory and computation cost).

## G.2   2D Denoising

Figure 4 shows 24 steps of denoising starting from a grid for the Swiss Roll gradient fields. At larger batch sizes and lower learning rates, and with more denoising steps (lower right panel), the MLP control begins to approximate the Swiss Roll data manifold.

## G.3   MNIST Infilling

Figure 5 shows the result of 3000 steps of denoising of MNIST train and test digits, after training for 1 epoch at batch size 1. This shows that the network, which has been trained on denoising small additive Gaussian noise perturbations to train set digits, is able to denoise unseen binary mask perturbations on unseen test set digits. This occurs over many iterative steps of denoising, much as the grid in Figure 4 is iteratively denoised to the Swiss Roll data manifold.

# H   Experimental Details

## H.1   UCI regression details

Each G-GLN was trained with batch size 1 for 40 epochs of a randomly selected 90% split of the dataset (except DO which was trained for 400). The predictive RMSE is evaluated for the remaining 10%, with the mean and standard error reported across 20 different splits (5 for Protein Structure). Similarly to [31], we normalize the input features and targets to have zero mean and unit variance during training. Target normalization is removed for evaluation.

For each UCI dataset we train a G-GLN with 12 layers of 256 neurons. Context functions are sampled as described in Section 4 with an additive bias of 0.05. The switching aggregation scheme was used to generate the output distribution. In [31] the authors specify that 30 configurations of learning rate, momentum and weight decay parameters are tuned for each task for VI, BP and PBP. We likewise search 12 configurations of learning rate $\in \{1e^{-3}, 3e^{-3}, 1e^{-2}\}$ and context dimension $\in \{4, 6, 8, 10\}$ for each task and present the best result.

Figure 4: G-GLNs (top set of rows) and MLPs (bottom two sets of rows) are trained on 1-step denoising of added Gaussian noise using data points sampled from a Swiss Roll. Subsequently, iterative multi-step denoising starting from a grid reconstructs an approximation of the original Swiss Roll data manifold. BS denotes batch size, LR denotes learning rate. The initial grid followed by 24 steps of denosing are shown left to right and top to bottom.

## H.2 SARCOS details

The G-GLN was trained for 2000 epochs using the SARCOS test and train splits defined in [35]. Inputs were normalized to have zero mean and unit variance during training, with the target component-wise linearly rescaled to $[-1, 1]$. Fixed bias Gaussians were placed with means $\pm 7$ and variance 5 along each of the 7 output coordinate axes. The network base model uses Gaussians with standard deviation 1 centered on each component $x_i$ of the input vector.

The G-GLN was trained with 4 layers of 50 neurons, context dimension 14, and learning rate 0.01. Context functions are sampled as described in Section 4 with an additive bias of 0.05. The switching aggregation scheme was used to generate the output distribution. We enforce weights to be in $[-10^5, 10^5]$ and mixed variances $\sigma^2_{\text{PoG}}$ to be in $[1, 10^9]$ by performing projections when needed.

## H.3 Contextual bandits details

We adopt the experimental configuration described in [1], including inputs and target scaling and method of hyperparameter selection. Performance was evaluated across 500 seeds per dataset. The G-GLN was trained with shape $[1000, 100, 1]$ with context dimension 1 and a learning rate of 0.003. A single output layer with a single neuron was used to generate the output distribution. Context functions are sampled as described in Section 4 with an additive bias of 0.05. For the GLCB algorithm a UCB exploration bonus of 1 was chosen with mean-based pseudo-count aggregation.

Figure 5: Further MNIST infilling examples. G-GLN was trained for 1 epoch at batch size 1 by denoising a small additive Gaussian noise pattern from each train image. Subsequently, it can remove unseen binary occulsion masks either from train images (left) or unseen test images (right). Orig: original image. Mask: masked image: Fill: filled image. Examples were randomly chosen.

### H.4 Denoising details

The MLP control for Swiss Roll denoising was a ReLU network with hidden layer sizes 64 and 32 and output size 3 (2D $\mu$ and 1D $\sigma^2$). Both were trained with Gaussian log likelihood. The MLP was evaluated with learning rates of both $0.01$ or $0.0005$ for comparison. For Hamiltonian Monte Carlo (HMC) sampling, 15000 HMC steps were performed, with each step consisting of 150 sub-steps and $\epsilon = 0.003$. No acceptance criterion was used. Particle mass was 1.

For the MNIST image denoising, the G-GLN was trained with 6 layers of batch size 50 with context dimension of 10 and a learning rate of 0.05. The network base model uses Gaussians with variance 0.3 centered on each component $x_i$ of the input vector. A single output layer with a single neuron was used to generate the output distribution.

For MNIST denoising, context functions are sampled as described in Section 4 with a normally distributed additive bias of scale 0.05, while for Swiss Roll denoising in 2D, the additive bias scale was 0.5 to ensure proper tiling of the low-dimensional input space with hyperplane regions.

The G-GLN was trained in a single pass through all train points with batch size 1, with data represented as flat $28^2 = 784$ dimensional vectors. The model was trained to remove a single additive Gaussian noise pattern for each train image during training, and was then tested on MNIST in-filling using an independent test set of images occluded by unseen randomly positioned binary masks. To estimate a gradient direction for infilling, a single step of the trained denoising procedure was performed on each successive image, then a step of length 0.002 was taken interpolating between the image and the denoised prediction, after which pixels outside the masked region were projected back to their original values. This was repeated iteratively up to 3000 times.

For both Swiss Roll and MNIST denoising, target data was component-wise linearly scaled to $[-1, 1]$. For MNIST, we first added Gaussian noise of standard deviation 75 to the first 10k train points to define an appropriate scaling range for the linear scaler. All weights were kept positive by clipping to a maximum of 1000. A minimum $\sigma^2$ was enforced by clipping during inference but not updating. Log-barriers were not used.