[Reviews · NeurIPS 2020]

Review 1

Summary and Contributions: The authors propose an extension of the GLN by modelling each neuron as a product of Gaussian. Doing so enables them to model regression tasks as well as density estimation tasks. Their G-GLN model is back-propagation free and updates directly using online gradient descent. Hence, the G-GLN is much more efficient in both computational resources and data dependency. Additionally, thanks to their model and loss function choices, their optimization problem is locally convex.

Strengths: 1. The GLN-based idea is very interesting and novel. Current SOTA models can achieve amazing results but require much more computational resources due to back-propagation. 2. The paper is very well written and easy to follow. The figures, tables, and algo-box are also easy to understand. 3. The paper provides sufficient background information on both prior work and model properties for the Gaussian distribution. 4. The paper shows competitive performance of the G-GLN model on various tasks such as regression, density estimation, and bandits.

Weaknesses: 1. For Table 2, on the SARCOS dataset, the G-GLN is trained with 1200 epochs. Are all models trained with the same number of epochs? 2. Similarly for Table 1, are all models trained with 40 epochs? I am not sure if the results for the G-GLN is the average prediction after 1 epoch or 40 epochs. More generally, I am curious if there is an advantage for the G-GLN model in terms of training epochs.

Correctness: The claims and method seem correct.

Clarity: The paper is very well written.

Relation to Prior Work: The paper provides sufficient background information and comparisons to its predecessor GLN model.

Reproducibility: Yes

Additional Feedback: Post rebuttal: keeping original score.


Review 2

Summary and Contributions: This paper introduces a new backpropagation-free deep learning algorithm for multivariate regression that leverages local convex optimization and data-dependent gating to model highly non-linear and heteroskedastic functions. Experiments have been conducted on several univariate and multivariate regression benchmarks in comparing with state-of-the-art. The proposed method outperforms competitive algorithms.

Strengths: The paper is well written and clear. The proposed framework G-GLN can be considered as an extension to the recently proposed GLN family of deep neural networks. In detail, authors extend the GLN framework from classification to multiple regression and density modelling by generalizing geometric mixing to a product of Gaussian densities. Many proofs of related theorems are given and experiments are sufficient.

Weaknesses: 1. Why “side information” is defined as the input features for an input example? Moreover, the function of “side information” should be further demonstrated by experiments. 2. The introduction of related work is not sufficient, and more work on GLN should be given to reflect the advantages or difference of the proposed method, such as the difference from B-GLN. 3. More detail experimental analysis should be connected with your objective function.

Correctness: Yes

Clarity: Good

Relation to Prior Work: Yes

Reproducibility: Yes

Additional Feedback:


Review 3

Summary and Contributions: The authors propose Gaussian gated linear networks, they extend the framework of GLN from classification to multiple regression. Their proposed approach is competitive on several benchmarks, including contextual bandits and density estimation.

Strengths: The authors build upon the work Bernoulli GLNs [2] and [3] and the well-known closure property of Gaussian distributions to extend GLNs to (multiple) regression and density estimation tasks, from which the concept of Gaussian GLNs arises naturally. The experiments show that G-GLN is competitive on several benchmarks covering (multiple) regression, online contextual bandits and density estimation.

Weaknesses: One could argue that the proposed approach is a straightforward extension of [2] and [3], thus limiting novelty. However, the connection to Gaussian distributions makes it an interesting approach. Some important details of the proposed approach are relegated to the supplementary material in favor of less interesting background material. The authors do not address the interpretability and robustness to catastrophic forgetting components of the proposed approach beyond mentioning them in the introduction.

Correctness: The technical details of the approach as well as the experimental setup seem correct and well-described.

Clarity: The paper is clearly written, the paper is well motivated and the assumptions, methodology and experiments are clearly stated.

Relation to Prior Work: The authors highlight the existing work on GLNs for classification and highlight that no prior work exists for regression tasks with GLNs.

Reproducibility: Yes

Additional Feedback: The authors may consider moving some of the details of the base model learning and switching aggregation to the main paper which are less obvious and more important to the proposed approach than well-known, textbook, properties of the Gaussian distribution (Section 2.1).


Review 4

Summary and Contributions: The paper proposes the Gaussian gate linear network (G-GLN) which it is an extension to the GLN family of deep neural networks. Properties of G-GLN are studied and authors examined G-GLN numerically using well know datasets and compared to other methods.

Strengths: The idea of GLN is local credit assignment mechanism by optimizing an objective and, previously the Bernoulli GLN was established. In this paper, authors uses the fact that exponential family densities are closed under multiplication to formulate G-GLN. It is demonstrated that the G-GLN is compatible through simulation studies.

Weaknesses: This is not my research area and I wasn’t able to grasp weakness. In the GLN, side information is fed to improve the learning instead of using all information. Could authors show or demonstrate the improvement due to side information? Also could author vaguely guide an optimal proportion for side information?

Correctness: The claims don’t seem to have any major fault.

Clarity: It was well structured and the idea, architecture, algorithm are well explained.

Relation to Prior Work: The relation to the previous work, Bernoulli GLN was stated.

Reproducibility: Yes

Additional Feedback:

[Author Response · NeurIPS 2020]

We thank the reviewers for their detailed and thoughtful comments and provide point-by-point responses below.

**=== Reviewer 1 ===**

**For Table 2, on the SARCOS dataset, the G-GLN is trained with 1200 epochs. Are all models trained with the same number of epochs?** The G-GLN was trained for 1200 epochs and all other models were trained for 54,000, as per original paper. We have since run the G-GLN for 2,000 epochs and achieved a state-of-the-art test MSE of 0.11.

**Similarly for Table 1, are all models trained for 40 epochs?** Yes, including the G-GLN.

**I am curious if there is an advantage for the G-GLN model in terms of training epochs.** G-GLNs are designed (e.g. convex local loss) to be more data efficient than contemporary methods and truncating results at 1 epoch still yields strong performance. These 1-epoch results will be added to the revision: Boston: 3.37 | Concrete: 7.33 | Energy: 2.33 | Kin8nm: 0.11 | Naval: 0.00 | Power: 3.92 | Protein: 4.38 | Wine: 0.63 | Yacht: 6.21

**=== Reviewer 2 ===**

**Why "side information" is defined as the input features for an input example?** The G-GLN architecture is very different from a standard MLP. In the case of a function $y = f(x)$, the input features "$x$" are input to the GLN as the side information. The side information is used for selecting active weights. The neuron uses the active weights to reweight the distributional predictions coming from the neurons in the previous layer (or "base predictions" in layer 0). This difference gives rise to the advantages of GLNs w.r.t. contemporary methods.

**The introduction of related work is not sufficient, and more work on GLN should be given to reflect the advantages or difference of the proposed method, such as the difference from B-GLN:** We agree that a more detailed treatment of B-GLNs is valuable for unfamiliar readers. This was pruned so that we could focus the 8 pages on our specific contributions, but we will use the additional page available to accepted papers to expand on connections to related work.

**More detail experimental analysis should be connected with your objective function.** We thank the reviewer for this comment but are unsure what they intend by experimental analysis. Our objective function is justified theoretically and underpins the success of the whole set of experimental results. In particular, using log-loss for density estimation is standard, but in our work it is foundational because it yields a closed-form representation for a product of exponential-family experts trainable via a convex loss function. This allows us to leverage the no-regret properties of online gradient descent to find a set of weights that gets increasingly close to the maximum likelihood solution as more data is seen.

**=== Reviewer 3 ===**

**One could argue that the proposed approach is a straightforward extension of [2] and [3].** We agree that, once the connection to a weighted product of experts and exponential family members is observed, it is straightforward to obtain an online learning formulation for a weighted product of Gaussians. However, it is certainly not obvious that this gives rise to a natural GLN formulation that works well in practice. This result should be of broad interest to the community, and we believe that the simplicity of the resulting algorithm is actually one of its key strengths given its strong performance on a broad number of tasks.

**Some important details of the proposed approach are relegated to the supplementary material.** This point is well-taken. Keeping a paper accessible yet interesting to the broadest cross-section of the community is a difficult trade-off, and we will certainly use the additional page of space available to accepted papers to upstream this material.

**The authors do not address ... interpretability and robustness to catastrophic forgetting.** Thank you for flagging this. These are properties that follow directly from the architecture and inductive biases of GLNs more generally (as has been shown in previous work on the B-GLN), but we could not find commonly accepted catastrophic forgetting / continual benchmarks for regression problems (beyond toy tasks like sine waves). We will revise the manuscript to clarify that the transfer of these properties to the G-GLN has not been shown in this work.

**=== Reviewer 4 ===**

No concerns were raised by Reviewer 4 other than to indicate that they would appreciate a more detailed assessment of broader impact. We thank the reviewer for their general comments and will include a more comprehensive impact statement in the revised submission, particularly with respect to the (positive) environmental and (potentially negative) privacy implications of improved data efficiency and online modeling.

[Meta-Review · NeurIPS 2020]

The paper proposes G-GLN, an extension to the recently proposed GLN family of deep neural networks. The proposed approach is competitive on several benchmarks, including contextual bandits and density estimation.